# Preparation and Characterization of Pure SiC Ceramics by HTPVT Induced by Seeding with SiC Nanoarrays

**DOI:** 10.3390/ma14216317

**Published:** 2021-10-22

**Authors:** Yu-Chen Deng, Nan-Long Zhang, Qiang Zhi, Bo Wang, Jian-Feng Yang

**Affiliations:** State Key Laboratory for Mechanical Behavior of Materials, Xi’an Jiaotong University, Xi’an 710049, China; dengyuchen@stu.xjtu.edu.cn (Y.-C.D.); zhangnanlong@stu.xjtu.edu.cn (N.-L.Z.); zhiqiang329@stu.xjtu.edu.cn (Q.Z.)

**Keywords:** silicon carbide, nanoarrays, HTPVT, PECVD

## Abstract

Dense SiC ceramics were fabricated by high-temperature physical vapor transport (HTPVT) growth process using SiC nanoarrays as the crystal seeds, which was obtained by vacuum heat treatment of amorphous SiC films prepared by plasma-enhanced chemical vapor deposition (PECVD) with a porous anodic aluminum oxide (AAO) template. In the HTPVT process, two-step holding was adopted, and the temperature at the first step was controlled at 2100 and 2150 °C to avoid SiC nanoarrays evaporation, and the grain size of SiC crystal increased with the increase in temperature and decrease in the pressure of Ar. The temperature of the second step was 2300 °C, and rapid SiC grain growth and gradual densification were achieved. The prepared SiC ceramics exhibited a relative density of more than 99%, an average grain size of about 100 μm, a preferred orientation along the (0 0 0 6) plane, a Vickers hardness of about 29 GPa, a flexural strength of about 360 MPa, and thermal conductivity at room temperature of more than 200 W·m^−1^·K^−1^.

## 1. Introduction

With the development of the semiconductor industry, ultra-pure substrate materials are required to apply at high temperature and processed with corrosive compounds. High-purity silicon carbide (SiC) materials, as a new generation of semiconductor material that can replace high-purity quartz and silicon, have become more and more important in advanced industries. In future fusion reactors, SiC materials are candidates for possible application as structural and functional materials; the main role is intended for the *blanket modules*. Because most SiC products are high value-added and have a broad market prospect, the research on high purity SiC materials is a focus at present [1,2,3,4].

Currently, the main methods of preparing dense SiC ceramics include recrystallization, reaction sintering, liquid phase sintering, spark plasma sintering, hot pressing, and hot isostatic sintering. The emphasis of various sintering methods is to improve the purity, density, and to reduce the size of grains to improve the mechanical properties of the SiC products. For the recrystallization method, silicon carbide green body is heated to a high temperature of more than 2000 ℃ to facilitate the evaporation–condensation. The resultant materials are porous due to no shrinkage [5], so they are mainly used as high-temperature components and diesel particulate filter materials [6]. The reaction sintering method is explored for the preparation of silicon carbide ceramics with low porosity. Silicon carbide powder with a certain particle size is used to reduce the porosity of the green body, and boron carbide is used as the carbon source to avoid polymer decomposition to form pores, so as to reduce the residual silicon [7]. Liquid phase sintering is mainly performed by adding Al_2_O_3_ and Y_2_O_3_ as sintering aids to react with the surface oxidation layer of silicon carbide at high temperature to form the liquid phase and promote the densification of the green body through liquid phase mass transfer [8,9]. High-density SiC ceramics can be prepared by hot pressing and hot isostatic sintering at a lower temperature, while this method is only suitable for preparing samples with small sizes [10,11]. Among these preparation methods, the recrystallization method can fabricate silicon carbide with high purity, heat shock resistance, and stable thermochemical properties. However, the density of recrystallized silicon carbide is relatively low, only about 2.5 g·cm^−3^, and the flexural strength is usually no higher than 140 MPa, which is not suitable for preparing dense materials [5]. The reaction sintering method can produce dense SiC ceramics at low temperatures of 1400~1600 ℃, but the residual silicon in the products means the obtained material can only be used at a temperature below 1370 ℃ [12]. For liquid phase sintering, hot pressing sintering, hot isostatic sintering, and spark plasma sintering, sintering aids are indispensable in the sintering process, which leads to the reduction in purity. Thus, the application of SiC materials is limited, especially in high-temperature and corrosion environments [13,14,15,16]. Therefore, it is significant to explore a method to prepare silicon carbide ceramics with high density and purity.

The high-temperature physical vapor transport (HTPVT) method is explored for the preparation of SiC single crystal [17,18]. The development of HTPVT is used to fabricate SiC ceramics, through nucleation and growth on the graphite substrate [19]. The HTPVT-grown SiC product is mainly used as a high-temperature and semiconductor component. Compared with the SiC grown by conventional CVD techniques, the SiC grown by the HTPVT method not only has a higher growth rate and larger sample size but also can be applied in a harsh environment, such as fusion reactors. The key to preparing SiC ceramic materials through HTPVT is the nucleation of SiC nuclei at the early growth stage, and the grain size of ceramics is determined by the number of nuclei. In the early growth stage of SiC ceramics fabricated by the HTPVT method, the reaction between Si, Si_2_C, and SiC_2_ in the vapor phase causes the formation of SiC nuclei on the graphite substrate. However, it is difficult to increase the number of nuclei, and the nucleation rate can only be increased by increasing the temperature. Due to the difficult random nucleation, the grain size of SiC ceramics is relatively large and the flexural strength of samples is relatively low.

In order to control the nucleation of materials, the introduction of nanoarrays as seeds is an effective method. Many methods have been reported to prepare nanoarrays, including electron beam lithography, nanoimprinting lithography, electrochemical micromachining, anodic alumina (AAO) template method, and so on. Most techniques can fabricate arrays with the nanoscale structure on substrates, however, too much machine-hour and high processing costs are indispensable [20]. Electron beam lithography has high precision, while the equipment is expensive and the machining area is small [21]. Nanoimprinting lithography and electrochemical micromachining have a slow machining speed, and the flexibility of adjusting the size of materials is insufficient [22,23]. Compared with all these methods, due to the advantage of high efficiency and low cost, the AAO template method can be used to prepare the nanomaterials with specific sizes and orientations [24].

Based on the previous research [25,26], in order to control the nucleation and optimize the microstructure of SiC ceramics more efficiently, we propose a new idea for preparing high-purity SiC ceramics by seeding with SiC nanoarrays. Firstly, using the porous ultra-thin AAO template, amorphous SiC films are prepared by the plasma-enhanced chemical vapor deposition (PECVD). Then, through vacuum heat treatment, the amorphous SiC films are crystallized and SiC ordered nanoarrays with small size and regular arrangement are obtained. Finally, the SiC nanoarrays are used as seeds for HTPVT growth to obtain SiC ceramics with grain size refinement, specific orientation, and good properties. Because the raw materials have a purity of 9.9% and any sintering aids such as Al_2_O_3_, Y_2_O_3_, and so on are not used in the HTPVT process, the obtained SiC ceramics have high purity.

## 2. Materials and Methods

The AAO template of 200 nm in length and 80 nm in diameter (Shanghai Shangmu Technology Co., Ltd., Shanghai, China) is transferred to the polished graphite substrate, and the amorphous SiC thin films with the regular arrangement are deposited in the holes of the AAO template by PECVD technology according to the deposition conditions as shown in Table 1. Then the AAO template is corroded and removed by the concentration of 1% NaOH solution, and the amorphous SiC films are retained. Finally, the crystalline SiC ordered nanoarrays are formed by heating the amorphous SiC films at 1200 °C in a high vacuum of less than 0.01 Pa

The commercially available SiC powders (mean particle size of 500 nm; purity of 99.9%, Shanghai ChaoWei Nanotechnology Co., Ltd., Shanghai, China) are used as raw materials. SiC powders are introduced in a medium-frequency vacuum induction furnace (3.5 kHz, Model ZGRS-160/2.55 Jinzhou Electric Furnace Co., Ltd., Jinzhou, China), together with the SiC ordered nanoarrays in a graphite crucible with an inner diameter of 50 mm. The temperature of the crucible is kept at 2100~2300 °C and the temperature difference between the source powders and the nano-array seeds is about 200 °C. Soaking time is 5 min~4 h and the pressure of Ar in the growth chamber is 2000~4000 Pa.

The chemical composition of SiC nanoarrays is characterized by X-ray photoelectron spectroscopy (XPS; ESCALAB Xi+, Thermo Fisher Scientific Inc., Waltham, MA, USA) that used Al as the target and had an energy step of 0.050 eV. The XPS process is performed at room temperature and high vacuum. Charging is corrected by setting the binding energy of the C_1s_ peak at 284.6 eV and samples are degassed in a vacuum oven overnight before XPS measurements. The phase formation of the SiC ceramics is identified by X-ray diffraction (XRD; X’Pert PRO, PANalytical Ltd., Almelo, The Netherlands) using Cu K*α* radiation on the polished sample. The XRD patterns are recorded in the 2*θ* range of 20–80°, with a step size of 0.01° and a scan speed of 10° min^−1^. The microstructure of samples is observed using a scanning electron microscopy system (S4800, Hitachi, Ltd., Tokyo, Japan). The average grain size is estimated through the Image-Pro Plus quantitative image analysis software (Version 7.0, Media Cybernetics Inc., Rockville, MD, USA). The densities of specimens are determined by the Archimedes method. Flexural strengths of specimens are tested via the three-point bending test (Model WDT-10, Tianshui Hongshan Test Machine Factory, Tianshui, China) with a support distance of 16 mm and a crosshead speed of 0.5 mm min^−1^. Vickers hardness is tested with a load of 9.8 N using the OmniMet88-7000 Fully Automated Microindentation Hardness Testing System. Thermal conductivity is tested by the laser thermal conductivity analyzer (Model LFA 467 HyperFlash, NETZSCH Group, Selb, Germany), and the direction of heat flow is along the c-axis of SiC samples (10 × 10 × 2 mm^3^).

## 3. Results and Discussion

Figure 1 shows the morphology of SiC nanoarrays at different stages before and after the PECVD process. Figure 1a shows a porous AAO template that transferred to the polished graphite, which has a large number of regular holes with a diameter of about 80 nm. Figure 1b shows the amorphous SiC films that form in and on the holes of the AAO template after the PECVD process according to the conditions shown in Table 1. By comparing Figure 1a with Figure 1b, it can be seen that the holes of the AAO template are filled and sealed by the amorphous SiC film. In addition, amorphous films deposited at the hole joints of the AAO template have a certain trend of transverse growth. Figure 1c shows the SiC nanoarrays obtained after removing the AAO template inside the sample with a 1% NaOH solution and heat-treatment at 1200 °C in a high vacuum of less than 0.01 Pa for 150 min. The structure of the nanoarrays is very regular, and the average size of the nano-array is about 50 nm, which is slightly smaller than the diameter of the AAO template.

Figure 2 presents the survey scan of XPS results. Si_1s_ peak and Si_2p_ peak of Si element, C_1s_ peak of C element, and O_1s_ peak of O element exist before and after heat treatment overall. Si and C elements are naturally present due to the use of gas sources and substrate material, and O elements are present due to the residual O of the AAO template. Through comparing Figure 2a and Figure 2b, it can be seen that the intensity of Si and O elements has barely changed, however, the intensity of C_1s_ peak is greatly enhanced by vacuum heat-treatment at 1200 °C for 150 min, which indicated that a part of C elements in the graphite substrate gradually spread into the SiC thin films, and reacted with the Si and O elements during the process of heat treatment.

Figure 3 shows the narrow scan of the XPS of the SiC nanoarrays after vacuum heat treatment. In Figure 3a, by comparing the standard binding energies of Si elements and fitting peaks, the narrow scan of Si_2p_ is divided into three components, corresponding to the binding energies of 101.3, 102.1, and 103.0 eV, respectively. Specifically, the peak corresponding to 101.3 eV can be considered as the Si–C bond, and its content is 35.73%. The binding energy of 103.0 eV can be considered as SiO_2_, namely the Si–O bond, and its content is 29.56%. The peak corresponding to 102.1 eV is an organic silicon compound, in which one organic group at least is directly connected to Si atoms. Based on the conditions of PECVD and heat treatment, this component is supposed to be siloxane with the bond of Si–O–Si. In the heat treatment process, part of SiO_2_ can be converted into siloxane at high temperatures. In Figure 3b, similarly, the narrow scan of C_1s_ also can divide into three peaks corresponding to the binding energy of 283.6 eV, 284.4 eV, and 285.4 eV. The structure with the lowest binding energy is the C–Si bond, which accounted for 10.82%. The peak corresponding to 284.4 eV can be regarded as the C–C bond, because of the use of graphite substrate. The C–O–C bond has the highest binding energy of 285.4 eV, with the highest content of 60.51%. In Figure 3c, the narrow scan of O_1s_ is only divided into two peaks. The peaks corresponding to 532.0 and 532.8 eV can be regarded as the C=O bond and SiO_2_, and the contents are 33.08 and 66.92%, respectively. These results indicate the bonding state and chemical composition of SiC nanoarrays after heat treatment. Si atoms and C atoms in the films and substrate mainly react with each other to become SiC, and a part of Si atoms and C atoms react with O atoms to form SiO_2_ and the structure of C–O–C, respectively.

In the SiC amorphous films prepared by PECVD, there are abundant Si–Si, SiH_n_ (n = 1, 2, 3), C–SiH, Si–C, and Si–CH_3_ free radicals. Through the XPS analysis, the crystallization process of SiC nanoarrays obtained by heat treatment at 1200 °C in a high vacuum is as follows: through heat treatment in high vacuum, the Si–H bond in SiH_n_ (n = 1, 2, 3), C–SiH, and Si–CH_3_ has broken, leading to the release of H atoms. The Si atoms with suspended bonds are released from Si–H and Si_1−x_C_x_ and react with the C atoms in the films, thus increasing the number of Si–Si and Si–C bonds. Meanwhile, C atoms on graphite substrate gradually diffuse in the films during heat treatment and react with Si atoms. Thus, the crystallized amorphous films and SiC ordered nanoarrays are obtained.

By analyzing the element composition and bonding state of the SiC nanoarrays, and estimating the integral strength of the narrow scan of Si_2p_ and C_1s_, the Si/C atomic ratio of the SiC nanoarrays can be further calculated. The ratio can be estimated by the following formula:(1)$$R_{\mathrm{Si}/C}=\frac{A_{X}S_{Y}}{A_{Y}S_{X}}$$
where X and Y represent the Si element and C element, respectively. *A*_X_ and *A*_Y_ represent the integral area of the narrow scan of Si_2p_ and C_1s_. *S*_X_ and *S*_Y_ are the sensitivity factor of the Si element and C element, respectively, with values of 0.205 and 0.17 [27,28]. In this way, according to the formula above, the Si/C atomic ratio of the SiC nanoarrays obtained in this study can be calculated to be about 0.95 ± 0.02, which is very close to the stoichiometric ratio of SiC. Although the actual Si/C atomic ratio has deviated from the stoichiometry ratio, the structural vacancies are not formed, because the excess C atoms may be solutionized into the SiC.

The SiC nanoarrays are introduced as seeds at the cold end of the growth system to study its evaporation at different temperatures without adding raw materials. Figure 4 shows the microstructure of SiC nanoarrays after a soaking time of 5 min at different temperatures. By comparison, it can clearly be seen that the number of SiC ordered nanoarrays decreases with the increase in temperature. At 2100 °C, SiC nanoarrays with a diameter of 50 nm can exist in large numbers. At 2150 °C, it can be seen that the SiC nanoarrays have a little evaporation, and the average grain size of the nanoarrays decreases to about 30 nm. However, there are still a considerable number of SiC nanoarrays, which can be used as crystal nuclei for inducing the growth of SiC ceramics. At 2200 °C, a large amount of SiC nanoarrays have evaporated, and the original graphite substrate is exposed. At 2250 °C, the SiC nanoarrays are nearly completely evaporated, which is very adverse to the seed-induced growth of SiC ceramics. Corresponding to Figure 4a–d, the activation energies can be calculated, and the values are 785.85, 754.67, 723.52, and 692.42 kJ·mol^−1^, respectively. It indicates that the activation energy decreases with the increase in temperature. The higher the temperature is, the easier the SiC nanoarrays are to decompose.

These results indicate that to ensure the role of SiC nanoarrays as seeds to induce growth efficiently, two-step temperature control should be explored. At the first step, the vapor phases containing Si and C are induced to grow along with the existing nano-array seeds, so the temperature of the raw materials should be controlled at 2100 or 2150 °C to avoid excessive evaporation of the nanoarrays. After a certain growth stage of SiC grains, the temperature of the raw material can increase to 2300 °C to accelerate the decomposition and sublimation of the raw materials, and the growth rate of ceramics is increased.

Figure 5 shows the morphology of SiC grains prepared by HTPVT for 10 min using nanoarrays as seeds at 2100 °C, and the pressure of Ar is 2000, 3000, and 4000 Pa, respectively. It can be seen that there are a large number of small columnar grains at this early stage, due to the significant increase in crystal nuclei induced by seeding. The columnar grains are grown along with the nano-array seeds and the average grain size decreases with the increase in the pressure of Ar. Because the vapor pressure on the surface of the nanoarrays is inhibited by the higher pressure of Ar, the nano-array seeds avoid a large amount of evaporation loss. The size of SiC grains grown by seed induction can be refined due to the stable existence of a large number of crystal nuclei. The average size of SiC grains obtained by seed induction for 10 min at 2100 °C with the Ar pressure of 2000, 3000, and 4000 Pa is 2.41, 2.09, and 1.96 μm, respectively.

Figure 6 shows the morphology of SiC grains prepared by HTPVT for 10 min using nanoarrays as seeds at 2150 °C with the Ar pressure of 2000, 3000, and 4000 Pa, respectively. Similar to Figure 4, it can be seen that there are a large number of nuclei in the early stage and the grain size also has a decreasing trend with the increase in Ar pressure. The average size of SiC grains obtained with these conditions is 2.95, 2.37, and 2.16 μm, respectively. The grain size prepared at 2150 °C is slightly larger than that at 2100 °C.

Figure 7 presents the average size variation of SiC grains grown by HTPVT for 10 min at 2100 and 2150 °C with the Ar pressure of 2000, 3000, and 4000 Pa, respectively. It can be seen that the average size of SiC grains increases with the increase in temperature and the decrease in the pressure of Ar at the early growth stage. At a lower temperature, the pressure of vapor components containing Si and C in the growth system is relatively low. With the increase in the growth temperature, the vapor pressure increases gradually, and the growth driving force of the SiC grains also increases. Therefore, the average grain size of SiC in the early stage increases with increasing temperature. On the other hand, with the increase in the pressure of Ar in the system, the vapor pressure of nanoarrays and the evaporation process are inhibited. A large number of seeds has been retained efficiently and the grains are further induced to grow along with the interface of the existing nano-array seeds. Therefore, the average grain size decreases with the increase in Ar pressure.

SiC grains have a certain stable growth after the first temperature control, and then the decomposition temperature of the raw materials is further set to 2300 °C to facilitate the growth of SiC ceramics. Figure 8 shows the morphology of SiC grains grown after two-step temperature control. It can be seen that the grain size of SiC ceramics increases to about 25 ± 5.50 μm rapidly after increasing the temperature, and SiC grains with a regular hexagonal shape have appeared.

Figure 9 is the top and side view of the morphology of dense SiC ceramics prepared by seed induced via HTPVT at 2300 °C for 4 h. The thickness of obtained SiC ceramics is 3.2~4 mm after 4 h growth, and the average growth rate is about 0.8~1 mm·h^−1^. In Figure 9a,c, it can be seen that with the growth for 4 h, the SiC ceramics have reached a high density of 3.187 ± 0.002 g·cm^−3^ and 3.182 ± 0.001 g·cm^−3^, respectively, and the grain morphology is regular. On the other hand, it can be inferred that the polycrystalline structure at this time has a certain degree of preferred orientation; the specific density and preferred orientation factor values are shown in Table 2. In Figure 9a,c, the average grain size of the samples increases to 94.21 and 106.95 μm, respectively. By comparing this with previous research [19], the grain size of SiC ceramics is further refined by using SiC ordered nanoarrays as crystal seeds, which indicates SiC nanoarrays play a very effective role in the structure optimization of SiC ceramics. Figure 9b,d are the side view of the morphology of SiC ceramics. It is easier and more comprehensive to observe the morphology of SiC ceramics. Through the side view, it can be seen that the grains of SiC ceramics induced by nano-array seeds have a more uniform columnar structure, and in the axial direction have a highly consistent preferred orientation; the specific grain orientation may be analyzed by XRD.

Figure 10a shows the X-ray diffraction pattern of the SiC grains of the sample obtained at 2100 and 2150 °C for 10 min within the Ar pressure of 4000 Pa. It can be seen that the crystallite of SiC obtained at the first step is 3C–SiC. Although the temperature of the raw material is 2100 and 2150 °C, however, the actual temperature of the deposition region at the cold end is only about 1900 and 1950 °C due to the existence of temperature gradient in the growth system. This temperature has not reached 2100 °C, corresponding to the temperature of the phase transformation from 3C–SiC to 6H–SiC.

With the decomposition temperature of 2300 °C in the raw materials, the temperature of the cold-end region also reaches 2100 °C according to the temperature gradient, which is corresponding to the phase transformation temperature from 3C–SiC to 6H–SiC. Figure 10b shows the X-ray diffraction pattern of the SiC bulk ceramics grown for 4 h at 2300 °C. Through the XRD analysis, the crystal type of the obtained SiC bulk ceramics is pure 6H-SiC. At 2*θ* = 35.6°; a very strong diffraction peak can be seen, corresponding to the 6H–SiC (0 0 0 6) plane. The Lotgering factors (*f*) to evaluate the preferred growth of SiC along the (0 0 0 *l*) planes can be calculated according to Equations (2) and (3)
(2)$$p=\frac{\sum I_{\left(00l\right)}}{\sum I_{\left(hkl\right)}}$$
where $\sum I_{\left(00l\right)}$ is the sum of the relative intensities of peaks corresponding to the (0 0 0 *l*) planes,$\;\sum I_{\left(hkl\right)}$ is the sum of the relative intensities of peaks corresponding to all planes. For absolutely randomly oriented grains, *p* is defined as *p*_0_ [29]. Therefore, the Lotgering factor (*f*) is defined as follows in Equation (3)


(3)
$$f=\frac{P-P_{0}}{1-P_{0}}$$


Through calculating for the SiC bulk ceramics obtained with the first step temperature control of 2100 and 2150 °C, and the second step temperature control of 2300 °C, the Lotgering factor along the (0 0 0 *l*) plane is 0.94 ± 0.01 and 0.91 ± 0.02, respectively. The results indicate that the SiC ceramics grown by seeding with SiC nanoarrays have a very high orientation along the (0 0 0 6) plane.

Table 2 presents the density, grain size, Lotgering factor, mechanical properties, and thermal conductivity at room temperature of SiC ceramics prepared by seeding with SiC nanoarrays via two-step temperature control in HTPVT. In the previous research [24,25], the relative density of samples was 97.8%, the flexural strength was only 309 ± 34 MPa, and the thermal conductivity was 199.68 W·m^−1^·K^−1^. Compared with those results, the performance of SiC ceramics induced by SiC ordered nanoarrays has been improved. Specifically, the relative density of the samples is more than 99.0%, which is very close to the theoretical density of SiC material (3.211 g·cm^−3^). With the decrease in the average grain size, the flexural strength (parallel to the growth direction) of SiC ceramics increases to 362 ± 35 MPa, and the Vickers hardness increases to 29.5 ± 0.4 GPa; these results indicate that the mechanical properties of SiC ceramics prepared via HTPVT can be improved significantly by the seeding with SiC nanoarrays. At the same time, the thermal conductivity at room temperature of SiC ceramics is more than 200 W·m^−1^·K^−1^, because of the high orientation along the growth direction of the (0 0 0 6) plane of SiC ceramics. These results indicate that not only are the mechanical properties of SiC ceramics improved greatly, but also the thermal conductivity at room temperature is still maintained at a high level.

Figure 11 shows the schematic diagram of the regulation mechanism of nano-array seeds on the microstructure of SiC ceramics during the growth process, by combining the HTPVT and PECVD, and the optimization of the structure and performance of high-purity SiC ceramics has been realized by the idea of seed-induced growth. First of all, the AAO template with a nano porous structure is transferred on a polished graphite substrate. Then SiC amorphous films are arranged in the regular holes of the AAO template through the PECVD process. Through removing the AAO and using vacuum heat-treatment, the reaction of Si atoms and C atoms leads to the increase in Si–Si bonds and Si–C bonds, thus the SiC ordered nanoarrays with specific nanostructure are crystallized. Next, the nanoarrays are introduced as seeds at the cold end of the growth system to induce the growth of SiC ceramics. The two-step temperature control is adopted to ensure the effective existence of the SiC initial grain and the subsequent growth of SiC ceramics. At the first step of lower temperature, the vapor phase including Si, Si_2_C, and SiC_2_, which are sublimated from the raw materials, can diffuse to the nano seeds under the action of the temperature gradient and concentration gradient. With the continuous deposition of vapor components along with the seeds, the growth of SiC grains with specific orientation proceeds gradually, and a certain number of the initial grains are obtained. With the increase in temperature, the growth rate and densification of SiC ceramics have been facilitated. Finally, the dense SiC ceramics with refinement structure are obtained. The obtained SiC ceramics not only have a small grain size but also have a highly preferred orientation. Meanwhile, through microstructure regulation of SiC nanoarrays on SiC ceramics, the properties of SiC ceramics have been improved.

## 4. Conclusions

Dense SiC ceramics are fabricated by the HTPVT method using SiC nanoarrays as the crystal seeds, which is obtained by vacuum heat treatment of amorphous SiC films prepared by the PECVD with porous AAO template.

During the process of the vacuum heat treatment, the SiC ordered nanoarrays are formed due to the reaction of Si atoms and C atoms in the amorphous films and substrate, leading to the increase in Si–C bonds. The nanoarrays are introduced as crystal seeds and two-step temperature control in HTPVT is explored. The nano-array seeds are stable and grow gradually at the first step of 2100 and 2150 °C, and the average size of SiC grains of the final bulk ceramics increases with the increase in temperature and decrease in the pressure of Ar. The rapid growth and the densification of SiC ceramics occur during the second step of 2300 °C.

Through microstructure regulation of SiC nanoarrays on SiC ceramics, the optimization of the structure has been realized and the properties of SiC ceramics have been improved. The obtained SiC samples exhibit a small grain size of about 100 μm, a highly preferred orientation along the (0 0 0 6) plane, a good flexural strength of about 360 MPa, a high Vickers hardness of about 29 GPa, and excellent thermal conductivity at room temperature of more than 200 W·m^−1^·K^−1^.

## Figures and Tables

**Figure 1 materials-14-06317-f001:**
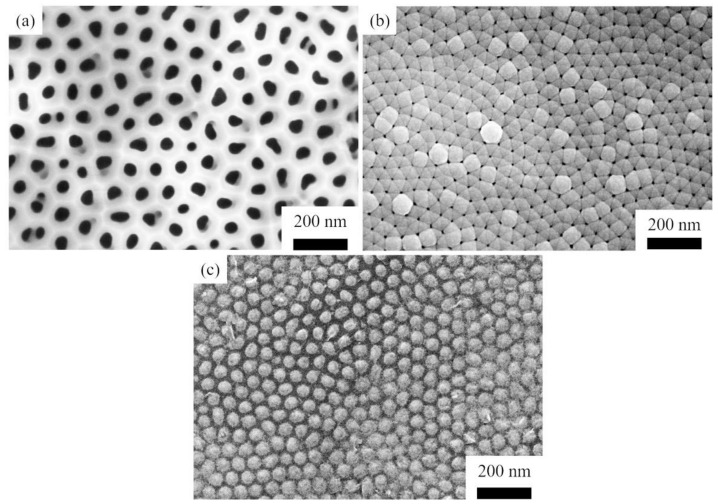
The morphology of SiC nanoarrays at each stage of the preparation process: (**a**) porous AAO template on graphite substrate; (**b**) amorphous SiC films in the holes of AAO; (**c**) SiC nanoarrays after removing AAO.

**Figure 2 materials-14-06317-f002:**
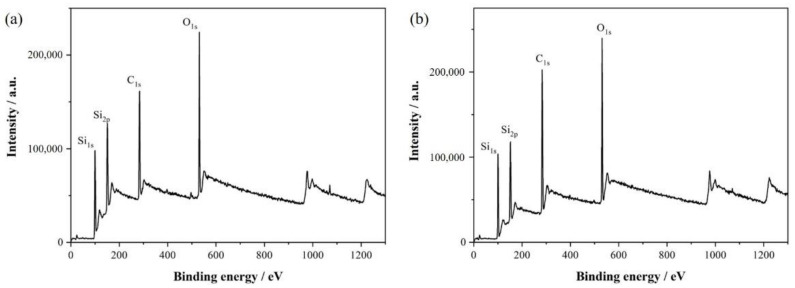
The survey scan of the XPS: (**a**) the SiC amorphous films obtained by removing the AAO; (**b**) the SiC nanoarrays obtained by vacuum heat-treatment.

**Figure 3 materials-14-06317-f003:**
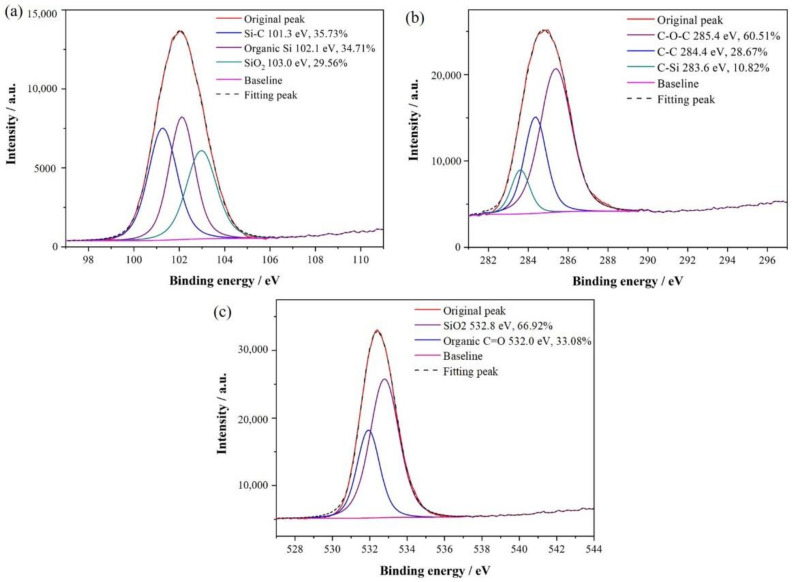
The narrow scan of the XPS of the SiC nanoarrays after vacuum heat-treatment: (**a**) Si_2p_; (**b**) C_1s_; (**c**) O_1s_.

**Figure 4 materials-14-06317-f004:**
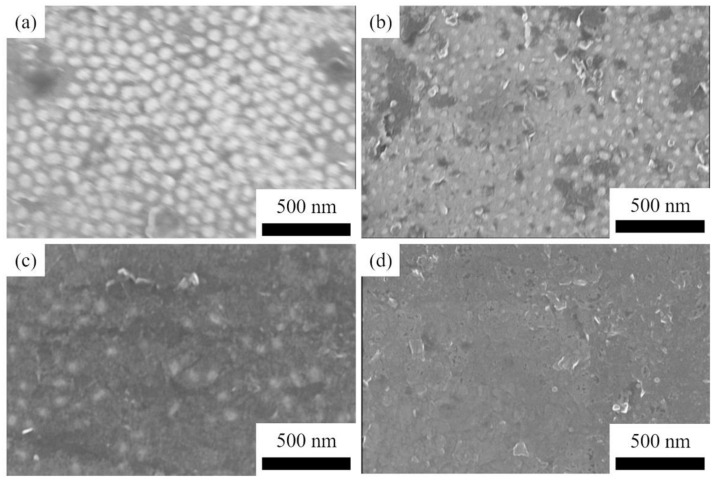
The microstructure of SiC nanoarrays after soaking time of 5 min at the temperature of the raw material from 2100 °C to 2250 °C: (**a**) 2100 °C; (**b**) 2150 °C; (**c**) 2200 °C; (**d**) 2250 °C.

**Figure 5 materials-14-06317-f005:**
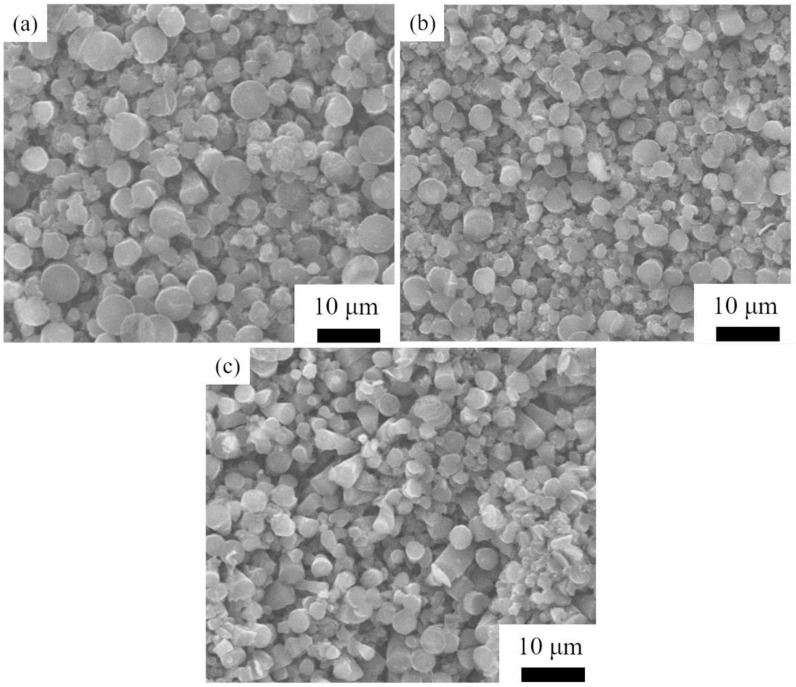
The morphology of SiC grains prepared by HTPVT at 2100 °C for 10 min: (**a**) 2000 Pa; (**b**) 3000 Pa; (**c**) 4000 Pa.

**Figure 6 materials-14-06317-f006:**
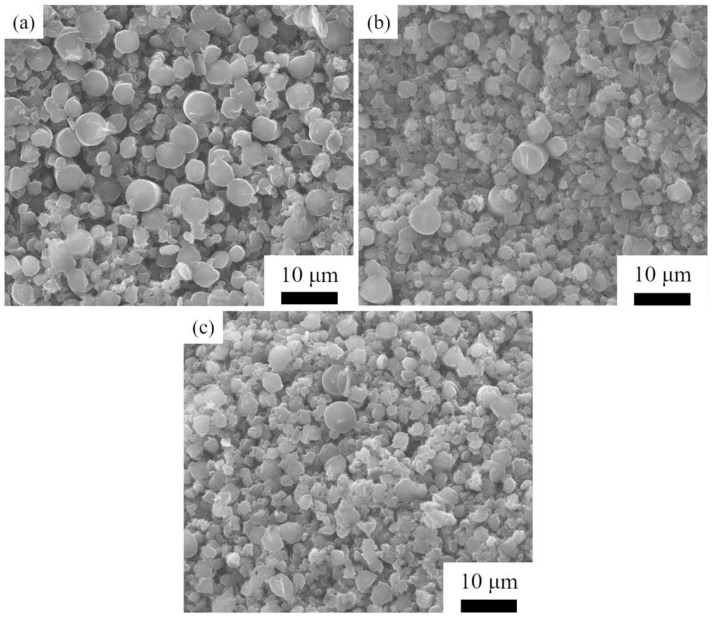
The morphology of SiC grains prepared by HTPVT at 2150 °C for 10 min: (**a**) 2000 Pa; (**b**) 3000 Pa; (**c**) 4000 Pa.

**Figure 7 materials-14-06317-f007:**
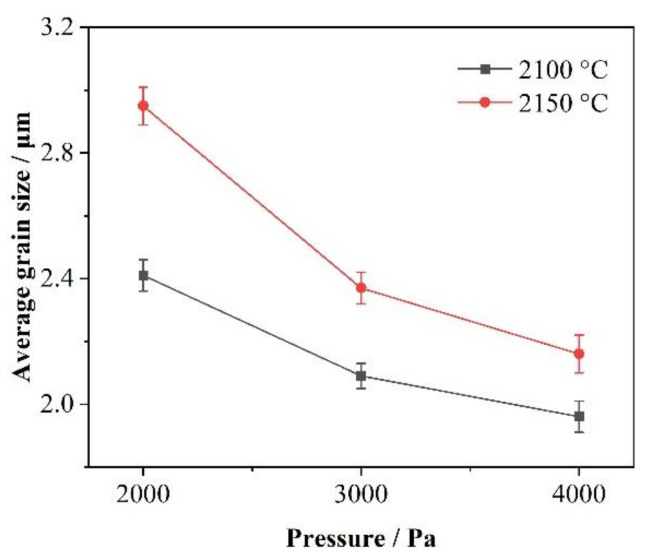
The relationship between the average grain size and the pressure of Ar at different temperatures.

**Figure 8 materials-14-06317-f008:**
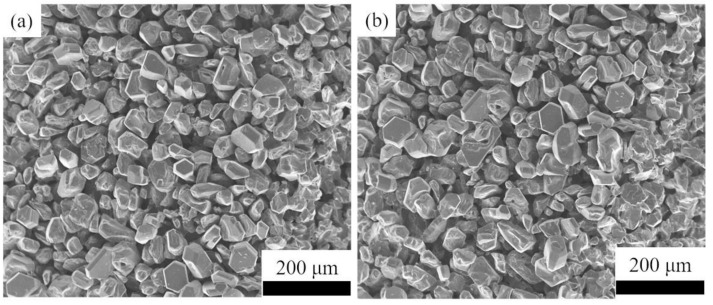
The morphology of SiC grains grown after two-step temperature control: (**a**) 2100 °C for 10 min and 2300 °C for 30 min; (**b**) 2150 °C for 10 min and 2300 °C for 30 min.

**Figure 9 materials-14-06317-f009:**
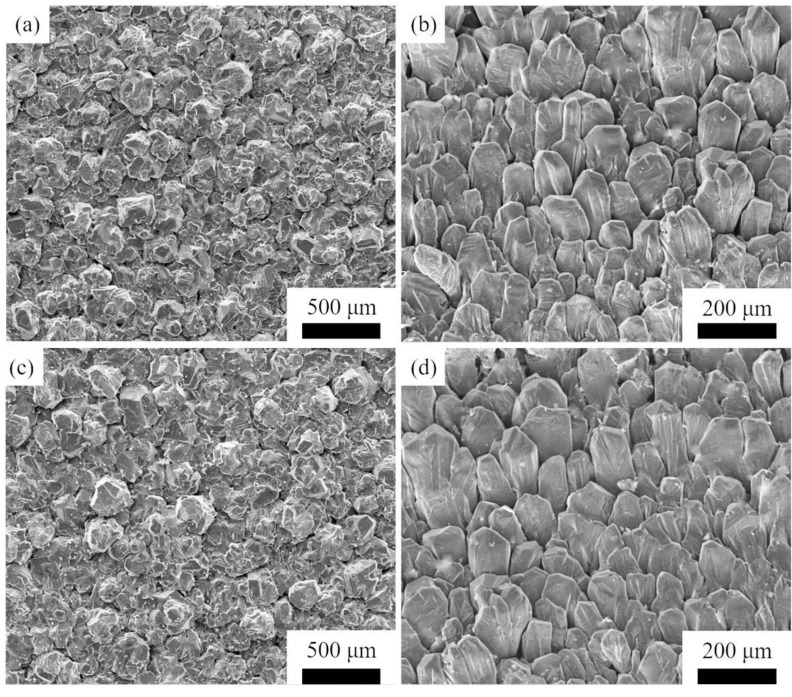
The morphology of dense SiC ceramics induced by nano-array seeds via HTPVT: (**a**,**b**) the top and side view of samples grown at 2100 °C for 10 min and 2300 °C for 4 h; (**c**,**d**) the top and side view of samples grown at 2150 °C for 10 min and 2300 °C for 4 h.

**Figure 10 materials-14-06317-f010:**
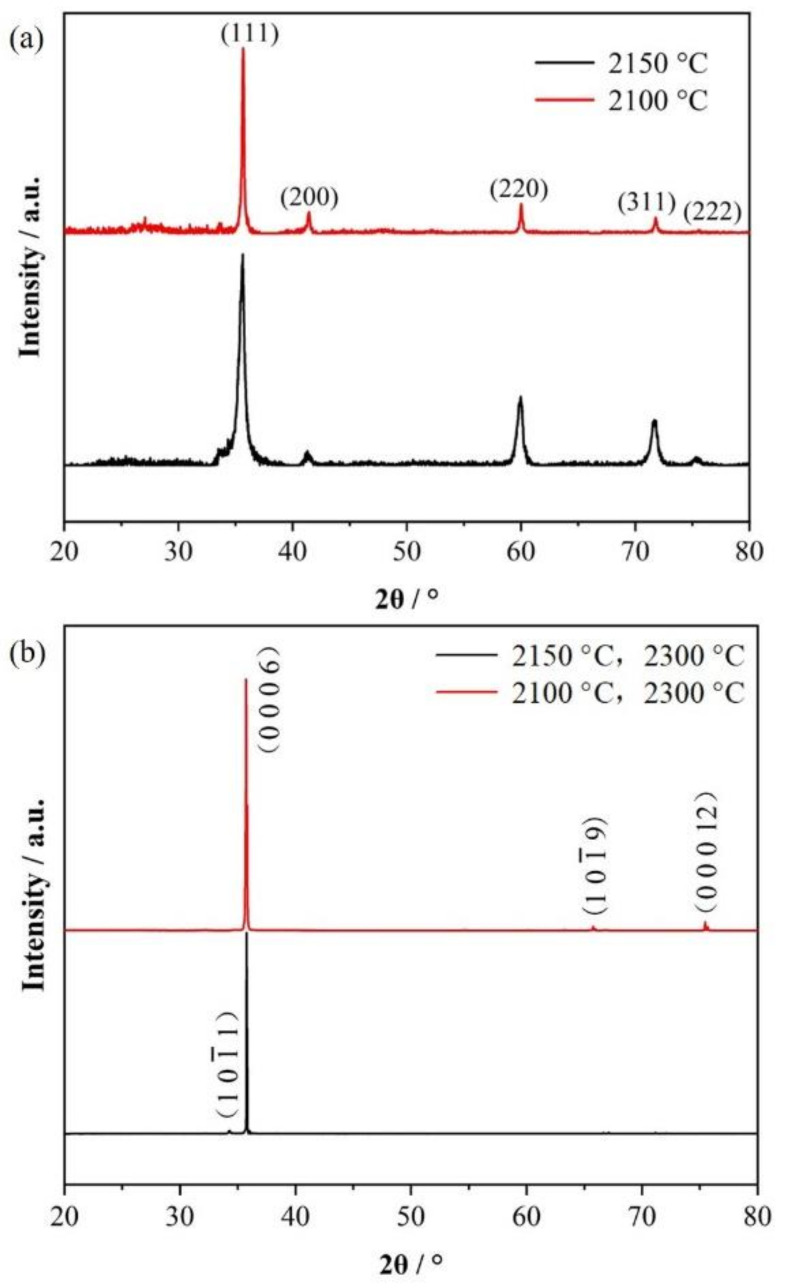
The XRD of the SiC samples: (**a**) grown at 2100 and 2150 °C for 10 min; (**b**) further grown at 2300 °C for 4 h.

**Figure 11 materials-14-06317-f011:**
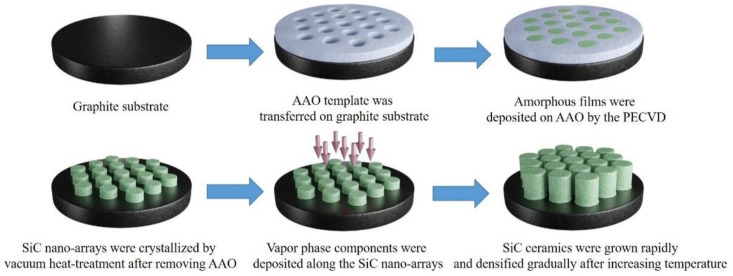
The schematic diagram of the regulation mechanism of nano-array seeds on the microstructure of SiC ceramics.

**Table 1 materials-14-06317-t001:** The deposition conditions of PECVD.

SiH_4_ Flow/Sccm	CH_4_ Flow/Sccm	H_2_ Flow/Sccm	Power/W	Temperature/°C	Pressure/Pa	Time/min
9	12	20	50	350	80	30

**Table 2 materials-14-06317-t002:** Performance of the SiC ceramics.

Two-Step Temperature Control/°C	Density/g·cm^−3^	Relative Density/%	Grain Size/μm	Lotgering Factor	Flexural Strength/MPa	Vickers Hardness/Gpa	Thermal Conductivity at RT/W·m^−1^·K^−1^
2100, 2300	3.187 ± 0.002	99.3 ± 0.01	94.21 ± 1.20	0.94 ± 0.01	362 ± 35	29.5 ± 0.4	213.42 ± 3.10
2150, 2300	3.182 ± 0.001	99.1 ± 0.01	106.95 ± 2.30	0.91 ± 0.02	347 ± 31	28.4 ± 0.7	202.15 ± 2.40

## Data Availability

The data presented in this study are available on request from the corresponding author.

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
