# Peer review of "Preparation and Characterization of Pure SiC Ceramics by HTPVT Induced by Seeding with SiC Nanoarrays"

_materials, 2021, doi:10.3390/ma14216317_

Round 1
Reviewer 1 Report
" In the Fig. 9 (a) and (c), it can be seen that with the growth for 4 hours, the SiC ceramics have reached a high density, and the grain morphology is regular." - please add the density values.
"Through microstructure regulation of SiC nano-arrays on SiC ceramics, the
optimization of the structure has been realized and the properties of SiC ceramics have been improved." - why do you believe that the columnar microstructure is optimal? Usually, it is associated with high anisotropy of mechanical properties and reduced corrosion resistance, so please provide additional argumentation for your statement. Also, please describe in what direction did you measure the flexural strength of the samples. I highly recommend providing the measurements for 2 orientations of the samples: parallel and perpendicular to the growth direction, in order to ascertain the mechanical anisotropy of the material.
Regarding the growth process - please provide the growth rate of the SiC specimens. I would also advise you to outline some use cases for the HTPVT-grown SiC and compare it to SiC grown by conventional CVD techniques.
There also are minor English errors, which should be corrected.
Reviewer 2 Report
In this paper, authors present a study of SiC ceramocs by HTPVP induced by seeding with SiC nanoarrays. The work is well focused, carried out and could be interesting for the Readers of Materials. However, some points should be considered for publishing.
Comments:
- SEM images throughout the work are too dark. The authors should modify them in order to the grains can be clearly distinguished.
- In the Introdutcion, define AAO acronymus in line 72 on page 2 instead in line 83.
- Include errors or desviations in all experimental data: for example in Figure 7 and Table 11 (including those for the relative densities).
- unify the number of decimals for same experimental data: for example some grain size are with two decimals and other ones with one decimal.
- Remove “35” from Flexural strength in table 11. The error is duplicated.
- Indicate the theoretical SiC density value used to calculate the relative density.
Reviewer 3 Report
This is a good article that could undoubtedly be published after some improvements and clarifications.
- The word "pure" in the title and below in the text should be explained in more detail. Do you mean free from impurities and defect-free? What concentration of point defects still allows the use of this word ?
- Continuing, it would be useful in the introduction to pay attention to research methods to determine how pure and defect-free SiC samples are. For example, in addition to XRD, Raman spectra and cathodoluminescence are always very useful for such purposes. See, for example, Harima, H. (2006). Raman scattering characterization on SiC. Microelectronic Engineering, 83(1), 126-129. Huczko, A., Dąbrowska, A., Savchyn, V., Popov, A. I., & Karbovnyk, I. (2009). Silicon carbide nanowires: synthesis and cathodoluminescence. physica status solidi (b), 246(11‐12), 2806-2808.
- It is also very important to note that SiC materials are very important for nuclear applications in strong radiation-induced environments, for example, for fusion technologies. See: Malo, M., Soto, C., García-Rosales, C., & Hernández, T. (2018). Stability of porous SiC based materials under relevant conditions of radiation and temperature. Journal of Nuclear Materials, 509, 54-61.
- Line 197. Is it correct that deviation from stoichiometry leads to the formation of structural vacancies?
- Fig. 4. Is it possible to determine the corresponding activation energies from the temperature dependences of the rearrangement of nanostructures ?
